# Prediction of Potential Suitable Distribution Areas for an Endangered Salamander in China

**DOI:** 10.3390/ani14091390

**Published:** 2024-05-06

**Authors:** Jiacheng Tao, Yifeng Hu, Jianping Jiang, Wanji Yang, Tian Zhao, Shengqi Su

**Affiliations:** 1College of Fisheries, Southwest University, Chongqing 400715, China; tcheng632@163.com (J.T.); hyf990312@163.com (Y.H.); 2CAS Key Laboratory of Mountain Ecological Restoration and Bioresource Utilization & Ecological Restoration Biodiversity Conservation Key Laboratory of Sichuan Province, Chengdu Institute of Biology, Chinese Academy of Sciences, Chengdu 610041, China; jiangjp@cib.ac.cn; 3Shengnongjia National Park Administration, Huibei Provincial Key Laboratory on Conservation Biology of the Shennongjia Golden Snub-Nosed Monkey, Shennongjia 442421, China; snjgjgy@126.com

**Keywords:** amphibian, maximum entropy, ENMeval, environmental factor, distribution pattern

## Abstract

**Simple Summary:**

Exploring species’ potential suitable habitats is crucial for endangered species conservation, in particular under future global climate change conditions. The Wushan salamander (*Liua shihi*) is an endangered salamander in China, which is a national protected species (level II). Based on the distribution records of *L. shihi*, the main objective of this study was to predict the distribution of suitable habitats under current and future climate conditions for *L. shihi*. Our results showed that precipitation, cloud density, vegetation type, and ultraviolet radiation were the main environmental factors affecting the distribution of suitable habitat for *L. shihi*. At present, the suitable habitats are mainly distributed in the Daba Mountain area. Under the future climate conditions, the area of suitable habitats increased, which mainly occurred in central Guizhou Province. These findings provided important information for the conservation of *L. shihi*.

**Abstract:**

Climate change has been considered to pose critical threats for wildlife. During the past decade, species distribution models were widely used to assess the effects of climate change on the distribution of species’ suitable habitats. Among all the vertebrates, amphibians are most vulnerable to climate change. This is especially true for salamanders, which possess some specific traits such as cutaneous respiration and low vagility. The Wushan salamander (*Liua shihi*) is a threatened and protected salamander in China, with its wild population decreasing continuously. The main objective of this study was to predict the distribution of suitable habitat for *L. shihi* using the ENMeval parameter-optimized MaxEnt model under current and future climate conditions. Our results showed that precipitation, cloud density, vegetation type, and ultraviolet radiation were the main environmental factors affecting the distribution of *L. shihi*. Currently, the suitable habitats for *L. shihi* are mainly concentrated in the Daba Mountains, including northeastern Chongqing and western Hubei Provinces. Under the future climate conditions, the area of suitable habitats increased, which mainly occurred in central Guizhou Province. This study provided important information for the conservation of *L. shihi*. Future studies can incorporate more species distribution models to better understand the effects of climate change on the distribution of *L. shihi*.

## 1. Introduction

Understanding the effects of human-induced perturbations on biological diversity is one of the central concerns in modern ecology [1]. During the past hundreds of years, human activities have dramatically changed the environment on Earth, in particular the climate, which has strongly affected animals in recent decades [2]. Based on previous studies, over 27% of mammals, 13% of birds, 21% of reptiles, 41% of amphibians, and 37%of fishes are threatened due to climate change and other human disturbances [3]. For instance, although the Atlantic Forest network of protected areas (PAs) supports 18% of the amphibians in South America, the number of amphibian species in PAs is declining under changing climate conditions [4]. The reduction of precipitation will lead to a decrease in the reproductive success rate of shovel-nosed frog (*Hemisus marmoratus*) and natterjack toad (*Bufo calamita*), resulting in a decrease in their populations [5]. Climate change can also cause the alteration of amphibian phylogenetic composition and niche. For instance, most of the amphibians in PAs contracted their ranges and such responses are clade specific. Basal amphibian clades (e.g., Gymnophiona and Pipidae) were positively affected by climate change, whereas late-divergent clades (e.g., Cycloramphidae, Centrolenidae, Eleutherodactylidae, Microhylidae) were severely impacted [6]. From the wet season to the dry season, the vertical niche space of amphibians in Sierra Llorona has a clear downward trend in response to natural levels of climate variability [7]. In recent decades, increasing studies also indicated that climate change can lead to the shift of animals’ geographical distribution. For instance, Nottingham et al. showed that the suitable habitats of Del Norte salamander (*Plethodon elongatus*) and Siskiyou Mountains salamander (*P. stormi*) will shift to the coast and out of the valley with a move north into the mountains under future climate change conditions in the Pacific Northwest of the United States [8]. Duan et al. demonstrated that amphibians in China would lose 20% of their original distribution ranges on average, and over 90% of species’ suitable habitats will shift to the north when compared with their current distribution range. As a consequence, climate change can lead to significant changes in the spatial pattern of amphibian diversity in China [9].

Among all the vertebrates, amphibians are particularly sensitive to climate change as they cannot regulate their body temperature actively [10]. This is especially true for salamanders, which possess some specific traits such as cutaneous respiration and low vagility [11]. However, studies focused on the effects of climate change on salamanders are still limited (but see [12,13]). The Wushan salamander (*Liua shihi*) is a national protected (level II) salamander in China, which was classified as Near Threatened in the Red List of China. Although this species was listed as Least Concern (LC) in the International Union for Conservation of Nature (IUCN), the wild populations have declined continuously in recent years [14,15]. Based on the records, this species is widely distributed in montane streams of eastern Sichuan, Chongqing, western Hubei, and southern Shaanxi Provinces, with the elevation ranging from 900 to 2350 m [16], and it mainly feeds on aquatic insects and algae [17]. In recent years, the wild population of *L. shihi* has been decreasing continuously due to human-induced perturbations [18]. Therefore, it is urgent to understand the distribution of suitable habitats of this species, as well as how the suitable habitat will shift under future climate change conditions.

Species distribution models (SDMs) have been proved to be effective to predict the effects of climate change on species distribution patterns [19]. Based on species distribution points and environmental data, these models predict where species likely inhabit using approaches such as statistical and machine learning analyses [20]. Accordingly, SDMs are involved in several models such as Bioclim, random forest, maximum entropy, regression tree, and genetic algorithm [21,22]. Although none of the above models can be regarded as the best one, the maximum entropy model (MaxEnt) was considered to exhibit higher prediction accuracy, have a stronger ability to integrate multiple environmental variables, and provide more intuitive results [23,24]. Therefore, MaxEnt is increasingly used in ecological studies to investigate the responses of species distribution patterns to climate change. Using this model, Zhao et al. demonstrated that climate change can induce different effects on the evolutionarily significant units (ESUs) of Chinese giant salamander (*Andrias davidianus*) in China, with the northern ESU exhibiting more severe habitat loss [25]. Moreover, Zank et al. used MaxEnt to investigate the potential effects of climate change on 24 species of red-bellied toads (*Melanophryniscus*) in South America, and they found that 40% of the species may lose over 50% of their potential distribution area by 2080 [26]. However, most studies only used the default parameters provided by the MaxEnt model, despite the fact that MaxEnt is sensitive to sampling bias and prone to overfitting when using default parameters [23,27]. Therefore, it is essential to optimize the parameters of the MaxEnt before conducting the model analyses [27].

The main objective of the present study was to assess the effects of climate change on the distribution of suitable habitats for *L. shihi*. Specifically, we (1) investigated the distribution of suitable habitats for *L. shihi* under current climate conditions; (2) analyzed the key environmental factors affecting the distribution patterns of *L. shihi*; (3) revealed the shift of suitable habitats (i.e., the distribution patterns and the area) caused by climate change in the future. Based on previous studies (e.g., [28,29]), we predicted that the suitable habitats of *L. shihi* are mainly distributed in southwestern China at present. We also predicted that climate change will lead to the expansion of suitable habitats from the current distribution area to the southwest. In addition, the area of suitable habitats would decrease due to climate change.

## 2. Materials and Methods

### 2.1. Study Area

*L. shihi* is an endemic amphibian species in China. Although its distribution records were concentrated in the Daba Mountains, its potential suitable habitats could be widely distributed in China. Therefore, and in order to better protect this endangered species, we considered the whole of China as the study area.

### 2.2. Species Occurrence Data

The occurrence data of *L. shihi* in this study were obtained from published literature (Appendix A), the Global Biodiversity Information Facility website (http://www.gbif.org, accessed on 28 May 2023) (Appendix A), and our original field survey (Appendix A). In total, 89 occurrence records of *L. shihi* were collected. To avoid spatial autocorrelation, redundant records within 5 × 5 km grids were excluded using SDMToolbox (version 2.4; [30]). Finally, a total of 53 occurrence records were obtained for further analyses (Figure 1).

### 2.3. Environmental Variables

Environmental variables were selected based on previous studies demonstrating that they may potentially affect the distribution of amphibians (e.g., [13,31,32]). These variables can be divided into five categories, including bioclimate, meteorology, vegetation, human disturbance, and topography. In total, we obtained 31 environmental variable raster layers (Table 1). Specifically, bioclimatic data were composed of 19 climate factors at a resolution of 2.5 min, which were derived from the WorldClim climate database (http://www.worldclim.org/, accessed on 26 May 2023) [33]. Meteorological factors were composed of ultraviolet-B (UV-B) radiation and cloud cover, which were derived from EarthEnv (https://www.earthenv.org/cloud, accessed on 3 June 2023) and Helmholtz Centre for Environmental Research (https://www.ufz.de/gluv, accessed on 3 June 2023), respectively. Vegetation data contained the percentages of tree coverage and vegetation types, which were from Global Map Data Archives (https://globalmaps.github.io/ptc.html, accessed on 3 June 2023) and Resources and Environmental Science Data Center (https://www.resdc.cn/, accessed on 3 June 2023), respectively. Human disturbance data were represented by the population density, which were downloaded from the Socioeconomic Data and Applications Centre (https://sedac.ciesin.columbia.edu/, accessed on 5 June 2023). Finally, topographic data included elevation, slope, and aspect at a resolution of 90 m, which were obtained from the Geospatial Data Cloud (https://www.gscloud.cn/, accessed on 5 June 2023). We unified their coordinate system as GCS_WGS_1984 and resampled them to obtain a consistent spatial resolution. 

In order to reduce the influence of spatial correlation, environmental variables with high correlation but low contribution rate were removed before the model analyses [34]. Correlation analysis was performed using SPSS26.0 software. A Shapiro test was conducted using R software version 4.3.2 (https://www.r-project.org/, accessed on 7 November 2023) before the correlation analysis to identify the distribution of each variable [35]. Variables with a normal distribution were tested by Pearson correlations, and others were tested using Spearman correlations [36]. For the contribution rate, we performed a pre-simulation test in MaxEnt 3.4.4 with the distribution data of *L. shihi* and the 31 environmental variables. The contribution rate of the variables was tested using the jackknife test [37]. After that, variables with too high correlations (|PCCs| ≥ 0.8; Figure 2) but a low contribution rate (<1%) were removed [38], and the rest of the variables were used for secondary simulation. Based on our results, 15 environmental variables were finally selected for constructing the final models, including five for bioclimate, five for meteorology, two for vegetation, one for human disturbance, and three for topography (Table 1).

The future climate data were obtained from the BCC-CSM2-MR climate system model [39]. These data contained two shared socioeconomic pathways (SSPs), SSP126 and SSP585, which are scenarios of global economic, demographic, and energy development in the future [40]. Specifically, SSP126 represents the combined effects of low vulnerability, mitigating stress, and radiative forcing. SSP585 represents the future socioeconomic path of high-emission, high-carbon (coal, oil, and natural gas) use [41]. In this study, two future climate scenarios (SSP126, SSP585) of three periods (2021–2040, 2041–2060, 2061–2080) were selected for projecting the future distribution area of *L. shihi*.

### 2.4. Parameter Optimization and Model Construction

There are five feature types in MaxEnt models, including linear (L), quadratic (Q), hinge (H), product (P), and threshold (T). For parameter adjustment, we computed the AICc values of the modeling parameters’ regularization multiplier (RM) and feature combination (FC; the combination of the above five feature types) in R software using the ENMeval package [42]. In this study, we considered the range of RMs from 0.5 to 4.0 and selected six FC types (i.e., L, LQ, H, LQH, LQHP, and LQHPT). Then, we used the parameters corresponding to the minimum information criterion AICc value to construct the species distribution models [27].

The distribution data, environmental variables, and the optimized model parameters were input into MaxEnt3.4.4 software (New York, NY, USA, https://biodiversityinformatics.amnh.org/open_source/maxent/, accessed on 17 May 2023). The importance of environmental variables to the distribution of *L. shihi* was evaluated according to the relative contributions of environmental variables and the results of the jackknife test [40]. Twenty-five percent of the distribution data were randomly selected as the test set, while the rest were considered as the training set. The maximum number of background points was 10,000. A total of 10 runs were set for model construction, and the replicated run type was cross-validation.

We used the receiver operating characteristic (ROC) curve and the area under the ROC curve (AUC) to evaluate the accuracy of the model. The range of the AUC values was 0–1. A larger value indicates higher model accuracy, as well as higher credibility of the model. Models can be considered as having high prediction accuracy when the AUC value is greater than 0.8, and then the prediction results can be adopted [43]. An AUC value greater than 0.9 indicates that the prediction accuracy of the model is extremely high [23].

### 2.5. Parameter Optimization and Model Construction

We imported the average value of MaxEnt output results into ArcGIS 10.8 software and used a conversion tool to convert layers from asc format to raster data. The habitat suitability degree was divided into four levels, including high suitability area, moderate suitability area, low suitability area, and unsuitable area by natural breaks (Jenks) [44]. Finally, we calculated the area and proportion of suitable areas for each level. Moreover, we analyzed the change trend from current to future scenarios.

## 3. Results

### 3.1. Model Optimization and Accuracy Evaluation

For the current distribution models, the ΔAICc exhibited the lowest value when feature combination (FC) = LQHP and regulation multiplier (RM) = 2.5, indicating that the model was optimal with these parameters (Figure 3). This best model showed that the AUC value of the working curve of the subjects was 0.992 ± 0.004 (mean ± standard deviation), indicating the extremely high accuracy of the model prediction, thus the overfitting phenomenon could be effectively avoided (Figure 4).

In terms of the future distribution models, the optimal parameters were FC = LQPH and RM = 2 for the SSP585 (2021–2040) scenario, while FC = LQ and RM = 0.5 for the rest of the scenarios. After applying the above parameters in MaxEnt to construct models, the results showed that the AUC values of the working curve of the subjects were all > 0.9.

### 3.2. The Importance of Environmental Variables

For the MaxEnt models constructed under the current climate scenario, the top five environmental variables accounted for 84.7% of the cumulative contribution, including precipitation of the driest month (Bio14, 29.7%), cloud cover seasonal concentration (Mseason, 28.6%), vegetation type (Veg, 15.8%), mean UV-B of the lowest month (UVB4, 5.6%), and slope (5%; Table 1). In terms of the permutation importance (the extent to which the model depends on the variable; [45]), the top five environmental variables were mean UV-B of the highest month (UVB3, 25.6%), precipitation seasonality (Bio15, 22%), mean temperature of the driest quarter (Bio9, 14%), mean UV-B of the lowest month (UVB4, 13.2%), and precipitation of the driest month (Bio14, 6.9%). For the jackknife test (Figure 5), the test gain value was 3.7 when considering all the environmental variables. When considering the variables individually, precipitation of the driest month (Bio14), vegetation type (Veg), precipitation seasonality (Bio15), cloud cover seasonal concentration (Mseason), and mean UV-B of the lowest month (UVB4) were the top five variables that exhibited the highest test gain values.

### 3.3. Current Potential Suitable Habitats for L. shihi

Based on the results of MaxEnt models (Figure 6, Table 2), the potential suitable habitat for *L. shihi* was widely distributed in southwestern China, including Chongqing, Hubei, Sichuan, Shaanxi, Hunan, and Guizhou Provinces. In total, the suitable distribution area under current climate conditions for *L. shihi* was 45.61 × 10^4^ km^2^. Specifically, the high-suitability region was mainly concentrated in the Daba Mountains and Shennongjia National Park, which are located at the junction of Chongqing, Hubei, and Shaanxi Provinces. In addition, there were a small number of high-suitability regions scattered in central Sichuan Province. The size of the high-suitability area was 6.51 × 10^4^ km^2^, accounting for 14.3% of the total suitable habitat. The moderate-suitability region included the eastern part of Sichuan, southern Shaanxi, western Hubei, and eastern Chongqing Provinces, showing a ring shape, and the area was 9.77 × 10^4^ km^2^, accounting for 21.4% of the total suitable habitats. The low-suitability region was wrapped around the periphery of the moderate- and high-suitability areas, showing a strip shape. Moreover, the area was 29.31 × 10^4^ km^2^, accounting for 64.3% of the total suitable habitats.

### 3.4. Future Distribution Patterns of the Suitable Habitats for L. shihi

In the 2021–2040 period under SSP126, the high-suitability area increased to 8.89 × 10^4^ km^2^, which was mainly contributed by the expansion in central Hubei Province and the junction of Chongqing and Guizhou Provinces. However, the high-suitability habitats in central Sichuan Province disappeared. The area of low-suitability habitats also decreased (244 km^2^), associated with the loss in central Sichuan Province. In the 2041–2060 period under SSP126, the total area of suitable habitats increased to 62.31 × 10^4^ km^2^. Specifically, more high-suitability habitats occurred in Guizhou Province, despite the concentrated area in the Daba Mountains decreasing. The moderate-suitability habitats in central Sichuan Province disappeared, while there was no obvious change for low-suitability regions. In the 2061–2080 period under SSP126, a continuous decrease in high-suitability region was observed in the Daba Mountains, with the area being about 8.14 × 10^4^ km^2^. The moderate-suitability region in Guizhou Province was lost, while the low-suitability region can be only observed in central and south China (e.g., Henan, Hubei, and Guizhou Provinces; Figure 7, Table 2). 

In the 2021–2040 period under SSP585, the high-suitability area increased to 8.16 × 10^4^ km^2^, which was mainly contributed by the expansion in the middle and north of Hubei Province. The moderate-suitability area expanded to the south, mainly located in the east of Chongqing Province and the north of Guizhou Province. The area of low-suitability habitats decreased (2.5 × 10^4^ km^2^), associated with the loss in the junction of Chongqing and Guizhou Provinces. In the 2041–2060 period under SSP585, the total area of suitable habitats increased to 53.50 × 10^4^ km^2^. Specifically, the high-suitability area expanded to Guizhou Province. The moderate-suitability area in the south of Henan Province expanded to 16.26 × 10^4^ km^2^, while the area of low-suitability habitats decreased (0.53 × 10^4^ km^2^). In the 2061–2080 period under SSP585, the highly suitable areas in the Daba Mountains were more concentrated. There was no obvious shift in the distribution pattern of the low-suitability area (Figure 7, Table 2). 

## 4. Discussion

In the present study, we used optimized MaxEnt models to predict the distribution patterns of suitable habitats for *L. shihi* in China under current and future climate conditions. Based on the high AUC values, our models can be considered to have high accuracy in prediction [46]. Many previous studies only used the default parameters when conducting MaxEnt models (e.g., [47,48]). However, the default parameters will lead to over-fitting and high omission rates of the model. The “ENMeval” package developed by Muscarella et al. based on the R language has been widely used for optimizing the regularization multiplier (RM) and feature combination (FC) in the MaxEnt model to balance the complexity and avoid those defects [42]. Recently, increasing numbers of researchers have argued that MaxEnt models should be optimized before conducting predictions, as the default parameters may cause some bias [27,49]. Our results supported this claim as we found that the types of FC and the values of RM could change in different models. However, more theoretical work and field work are still needed to verify the effectiveness of parameter optimization in MaxEnt models.

As ectothermic animals, amphibians’ growth and distribution are strongly affected by external environments, in particular the climate conditions [28,29]. This is especially true for salamanders, which are more sensitive to the change in climatic factors [13]. Among all the climatic variables, precipitation of the driest month was the most important one that determined the distribution of potential suitable habitats for *L. shihi.* Based on previous studies [17], the breeding period for this species is between March and April, associated with the dry season in the Daba Mountains. Therefore, sufficient precipitation can provide suitable spawning sites for *L. shihi* in montane streams, and permanent streams were critical for them to complete the life cycle [50]. High concentration of cloud cover, woody plant coverage, and low UV-B also were the main environmental variables that affect the distribution of suitable habitats for *L. shihi*. This is consistent with previous studies showing that ultraviolet light can cause oxidative stress, DNA damage, and egg death in salamanders [51,52]. Since an increase in ultraviolet rays may also lead to dramatic habitat reduction and connectivity fragmentation in other amphibian species that live in montane streams (e.g., spiny-bellied frog: *Quasipaa boulengeri*; [31]), low UV-B could be an important factor driving the survival and distribution of aquatic amphibians. In the present study, it was found that high cloud density and forest coverage can effectively reduce the damage of ultraviolet rays to *L. shihi* [53], supporting the survival and distribution of this species. In addition, the influence of slope cannot be ignored, which was associated with the water flow rate and sunshine angle of the habitat, and salamanders usually preferred to select places with low water flow rate and sufficient light to grow and reproduce [54,55].

Our results showed that the high-suitability habitat for *L. shihi* was concentrated in the junction of Chongqing, Shaanxi, and Hebei Provinces, suggesting that this species may prefer some specific ecological conditions in this area [56]. Therefore, this region should be paid more attention for the protection of this species. For protected animals, a concentrated distribution pattern means they may be more easily threatened by climate change, and regional natural disasters and disease transmission will put the entire population at risk of extinction [57]. Interestingly, there were no distribution points recorded in some high-suitability regions (e.g., central Sichuan and central Chongqing Provinces), indicating that further field investigations can be carried out in these areas. In addition, a small number of existing distribution points were located in low-suitability or even non-suitable areas, suggesting that these populations should be paid more attention.

In the future, the total area of the suitable habitats for *L. shihi* will increase, although the main spatial distribution patterns did not change dramatically. This may be due to the unique climatic conditions (cool and humid all year round) in the Daba Mountains and Shennongjia National Park, which are climate transition regions between subtropical and northern warm temperate zones [58,59]. In two periods (2021–2040 and 2061–2080), the area of the suitable habitats under SSP126 was smaller than that of SSP585. This shows that the high-emission and high-carbon use scenario (SSP585) may cause an increase in the area of suitable habitat for *L. shihi*, which is similar to the finding of Wider et al. showing that the suitable range of the blue-spotted salamander (*Ambystoma laterale*) and the red-backed salamander (*P. cinereus*) increases with the increase in greenhouse gas concentration [60]. From the time point of view, the suitable area of *L. shihi* in the future is larger than the current results, and the high-suitability area under the SSP585 scenario will gradually increase with time. This increase may be the cumulative effect of climate change. This is contrary to previous studies showing that the area of suitable habitats of some other salamanders (e.g., leprous false brook salamander: *Pseudoeurycea leprosa,* streamside salamander: *A. barbouri*, and Cheat Mountain salamander: *P. nettingi*) will significantly decrease in the future [45,61]. We speculated that under this scenario, climate change has just reached the suitable conditions for *L. shihi* in some areas. It is worth noting that in the next three periods, the distribution range will be more concentrated. It indicates that the concentrated areas may have more important protection significance, as this region should be the refuge for *L. shihi* under future climate change.

## 5. Conclusions

In conclusion, the present study predicted the potential suitable habitats for *L. shihi* using a MaxEnt model with optimized parameters under current and future climate change scenarios for three time periods (SSP126 and SSP585). Our results indicated that precipitation of the driest month (Bio14), cloud cover seasonal concentration (Mseason), vegetation type (Veg), mean UV-B of the lowest month (UVB4), and slope are important environmental variables that have a great impact on the habitat suitability. The suitable habitats under the current situation are mainly distributed at the junction of Chongqing, Shaanxi, and Hubei Provinces. Under future climatic conditions, the total suitable area increased. The new suitable habitats were concentrated in the central part of Guizhou and Hubei Provinces. However, suitable habitats located in the central part of Sichuan and Chongqing Provinces were lost. The results of this study can help us better understand the distribution of *L. shihi* and can provide important information for determining the suitable areas of this species in China. Future studies can incorporate more species distribution models to better understand the effects of climate change on the distribution of suitable habitats for *L. shihi*.

## Figures and Tables

**Figure 1 animals-14-01390-f001:**
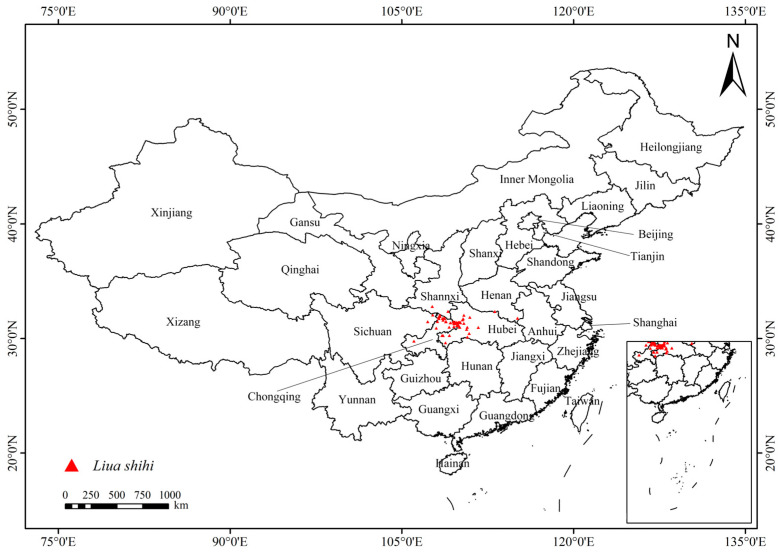
Distribution points of *L. shihi* (red triangles; after excluding autocorrelation).

**Figure 2 animals-14-01390-f002:**
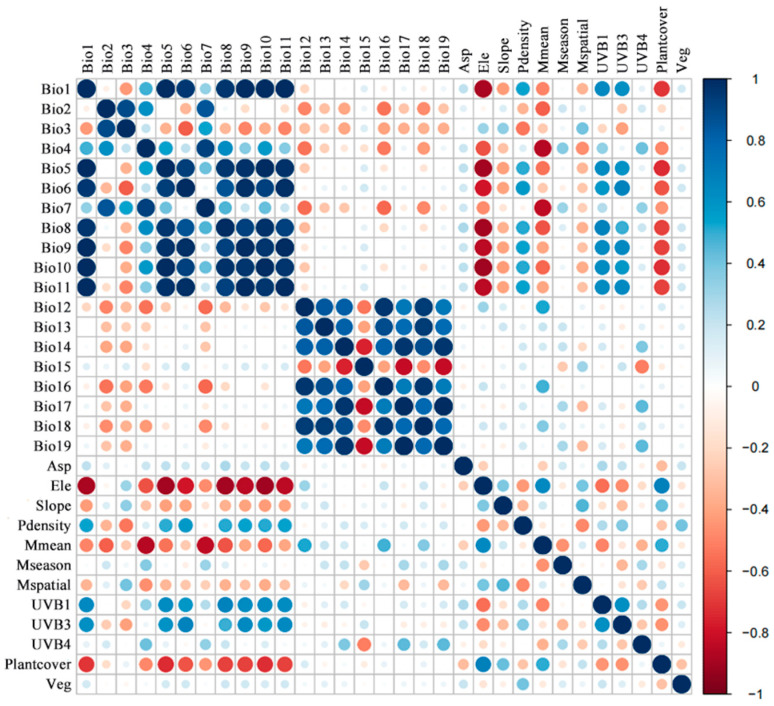
Correlation matrix between environmental variables. Bio1–19 are bioclimatic variables obtained from WorldClim website. Asp: aspect; Ele: elevation; Slope: slope; Pdensity: density of population; Mmean: mean annual cloud cover; Mseason: cloud cover seasonal concentration; Mspatial: cloud cover spatial variability; UVB1: annual mean UV-B; UVB3: mean UV-B of highest month; UVB4: mean UV-B of lowest month; Plantcover: density of trees on the ground; Veg: vegetation type. Positive correlations are displayed in blue and negative correlations in a red color. The color intensity and the size of the circle are proportional to the correlation coefficients.

**Figure 3 animals-14-01390-f003:**
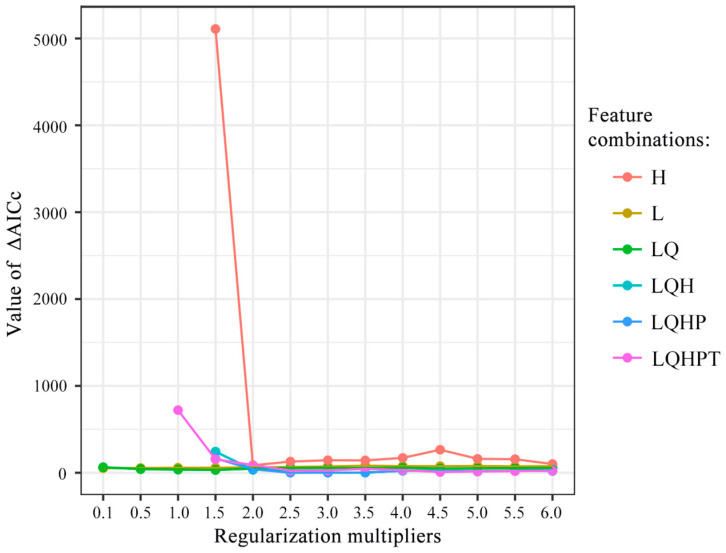
ΔAICc of the MaxEnt models under different regularization multipliers (RMs) and feature combinations (FCs).

**Figure 4 animals-14-01390-f004:**
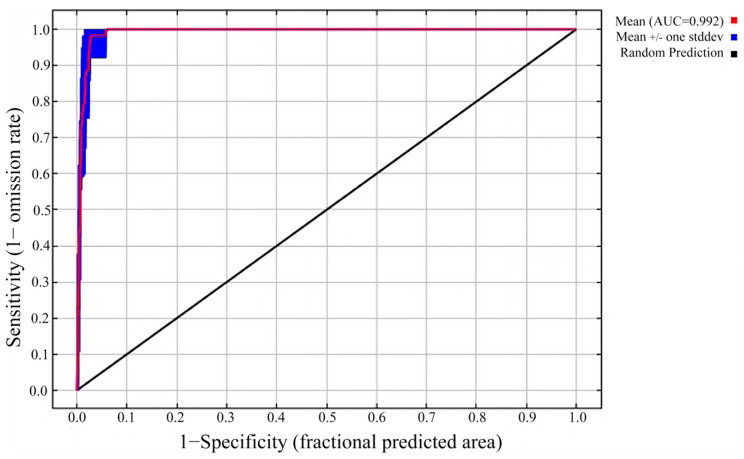
Receiver operating characteristic (ROC) curve and AUC value.

**Figure 5 animals-14-01390-f005:**
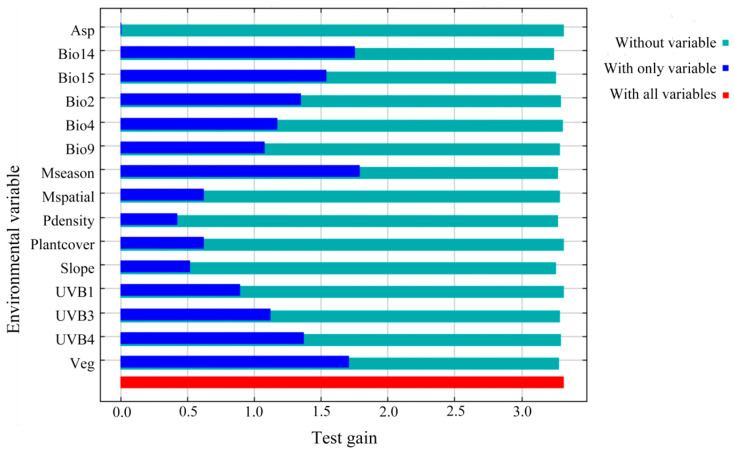
Jackknife of test gain for environmental variables in *L. shihi*.

**Figure 6 animals-14-01390-f006:**
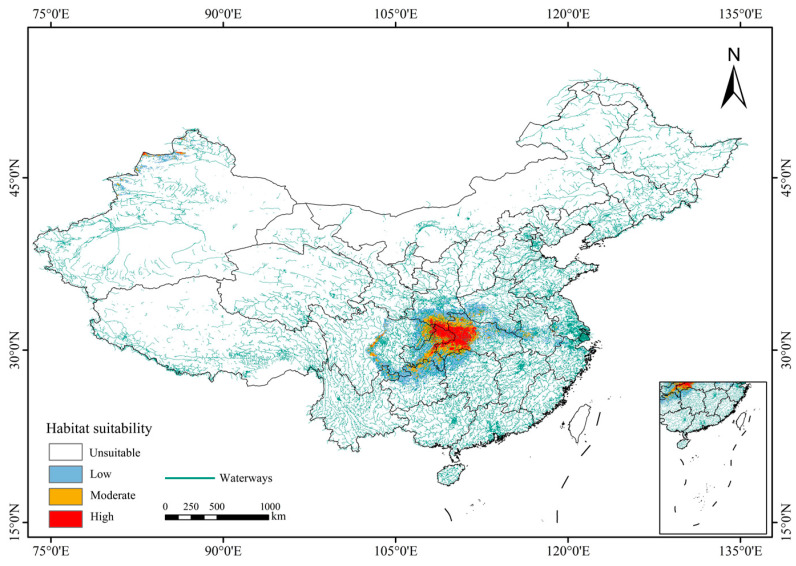
Potential suitable habitat for *L*. *shihi* under current climatic conditions.

**Figure 7 animals-14-01390-f007:**
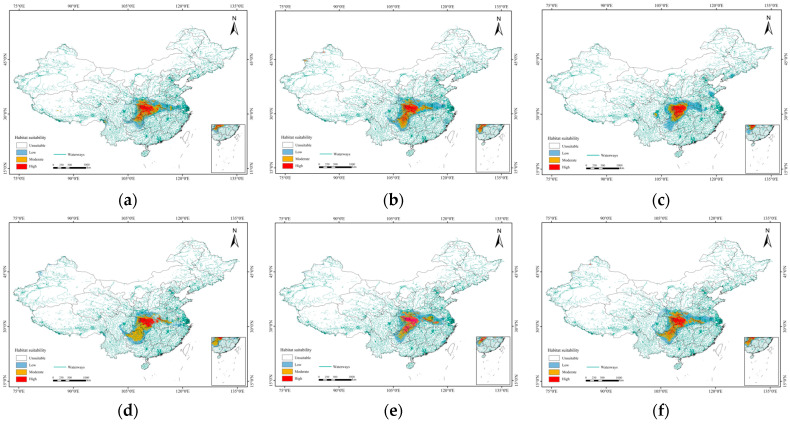
Potentially suitable climatic distribution of *L. shihi* under different climate change scenarios: (**a**) SSP126 from 2021–2040; (**b**) SSP126 from 2041–2060; (**c**) SSP126 from 2061–2082; (**d**) SSP585 from 2021–2040; (**e**) SSP585 from 2041–2060; (**f**) SSP585 from 2061–2080.

**Table 1 animals-14-01390-t001:** Contribution and permutation importance of environmental variables in MaxEnt models.

Code	Envirnonmental Variable	Percentage Contribution (%)	Permutation Importance (%)
Bio14	Precipitation of driest month	29.7	6.9
Mseason	Cloud cover seasonal concentration	28.6	6.6
Veg	Vegetation type	15.8	0.5
UVB4	Mean UV-B of lowest month	5.6	13.2
Slope	Slope	5	5
Bio2	Mean diurnal range	3.5	0.6
Mspatial	Cloud cover spatial variability	3.5	1.6
Bio9	Mean temperature of driest quarter	3.1	14
Bio15	Precipitation seasonality	1.6	22
Bio4	Temperature seasonality	1.4	2.5
UVB3	Mean UV-B of highest month	1.2	25.6
Pdensity	Density of population	0.4	1.5
UVB1	Annual mean UV-B	0.4	0
Asp	Aspect	0	0.1
Plantcover	Density of trees on the ground	0	0

**Table 2 animals-14-01390-t002:** The area of suitable habitats in different periods (values in the brackets indicate the variations when compared with the area of the current period) (×10^4^ km^2^).

Grade	Current	2021–2040		2041–2060		2061–2080	
		SSP126	SSP585	SSP126	SSP585	SSP126	SSP585
Low	29.306	29.281 (−0.025)	26.800 (−2.506)	36.037 (6.731)	28.774 (−0.532)	30.802 (1.496)	35.229 (5.923)
Moderate	9.774	16.646 (6.872)	20.535 (10.761)	17.336 (7.562)	16.262 (6.488)	9.889 (0.115)	19.076 (9.302)
High	6.526	8.887 (2.361)	8.162 (1.636)	8.931 (−2.405)	8.457 (1.931)	8.137 (1.611)	9.307 (2.781)
Total	45.609	54.814 (9.205)	55.497 (9.888)	62.306 (16.697)	53.493 (7.884)	48.828 (3.219)	63.613 (18.004)

## Data Availability

The datasets presented in this study are available in Appendix A.

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
