# Peer review of "Prediction of Potential Suitable Distribution Areas for an Endangered Salamander in China"

_animals, 2024, doi:10.3390/ani14091390_

Round 1

Reviewer 1 Report

Comments and Suggestions for Authors

Comments:

The manuscript, which focuses on forecasting potential habitat areas for an endangered salamander species in China, presents compelling empirical data and offers a novel perspective. The effort to devise strategies for the conservation of a species like Liua shihi is commendable. Nonetheless, certain revisions are necessary before it can be considered for publication.

The introductory section requires significant revision as it currently lacks relevance to the specific topic at hand. To effectively highlight the severe impacts of climate change, it is advisable to include more examples and data specifically related to amphibians, rather than other classes of species. The methodology description seems out of place in the introduction; it should be condensed and placed towards the end of the introduction, following the discussion of the species of concern. Additionally, the introduction lacks crucial information about L.shihi, including details about its habitat, distribution, global population trends, and IUCN status. These points should be addressed and incorporated into the revised version of the introduction.

The manuscript is deficient in providing information about the study area. Additionally, the title may be misleading as it does not clearly indicate whether the study pertains to the entire country of China or a specific region within it. It is necessary to include a detailed description of the study areas to address these concerns.

Comments on the Quality of English Language

Minor editing required 

Author Response

Reviewer 1:

The manuscript, which focuses on forecasting potential habitat areas for an endangered salamander species in China, presents compelling empirical data and offers a novel perspective. The effort to devise strategies for the conservation of a species like Liua shihi is commendable. Nonetheless, certain revisions are necessary before it can be considered for publication.

Reply: We appreciate the general interests of Reviewer 1 for our study. We have carefully and thoroughly revised this manuscript following Reviewers comments. Details have been provided below:

The introductory section requires significant revision as it currently lacks relevance to the specific topic at hand. To effectively highlight the severe impacts of climate change, it is advisable to include more examples and data specifically related to amphibians, rather than other classes of species. The methodology description seems out of place in the introduction; it should be condensed and placed towards the end of the introduction, following the discussion of the species of concern. Additionally, the introduction lacks crucial information about L.shihi, including details about its habitat, distribution, global population trends, and IUCN status. These points should be addressed and incorporated into the revised version of the introduction.

Reply: First, we have provided more focus on the examples of climate change on amphibians in the revised manuscript. Second, the methodology description part has been moved after the description of the target species (L. 85-105). Finally, more details of L. shihi have been provided (L. 70-84).

The manuscript is deficient in providing information about the study area. Additionally, the title may be misleading as it does not clearly indicate whether the study pertains to the entire country of China or a specific region within it. It is necessary to include a detailed description of the study areas to address these concerns.

Reply: The MaxEnt model was conducted at the scale of whole China. Following Reviewer 1’s suggestions, the study area section has been provided in the revised manuscript (L. 117-121).

Reviewer 2 Report

Comments and Suggestions for Authors

I do not have any major issues with this paper. 

In the first paragraph of discussion lines 277-286, it would be helpful to explain how the parameters for Maxent cane optimised, and what the rationale for such optimisation is.

minor points:

please place common names with genus/species names throughout, given that examples from a variety of animal groups are mentioned.

line 301-302: 'we argue the existence of low UV-B driven of aquatic amphibians living and distribution' - please clarify

line325 'high radiative forcing scenario - please clarify

please proof-read references carefully, and place genus/species names in italic font.

table 2, 'grade' is better as low/moderate/high

Comments on the Quality of English Language

moderate text editing is needed to remove typos and minor grammar errors

Author Response

Reviewer 2:

I do not have any major issues with this paper.

In the first paragraph of discussion lines 277-286, it would be helpful to explain how the parameters for Maxent cane optimised, and what the rationale for such optimisation is.

Reply: Done in the revised manuscript (L. 306-310).

minor points:

please place common names with genus/species names throughout, given that examples from a variety of animal groups are mentioned.

Reply: Done throughout the whole manuscript (e.g., L. 51 & L. 62).

line 301-302: 'we argue the existence of low UV-B driven of aquatic amphibians living and distribution' - please clarify

Reply: This sentence has been revised to make it clearer (L. 331).

line325 'high radiative forcing scenario - please clarify

Reply: Done (L. 356).

please proof-read references carefully, and place genus/species names in italic font.

Reply: Done throughout the whole manuscript.

table 2, 'grade' is better as low/moderate/high

Reply: Done (L. 270).

moderate text editing is needed to remove typos and minor grammar errors

Reply: We have carefully checked the language throughout the whole manuscript.

Reviewer 3 Report

Comments and Suggestions for Authors

I have read the manuscript entitled “Prediction of potential suitable distribution areas for an endangered salamander in China” by Jiacheng Tao and colleagues, submitted to the Animals magazine. The manuscript presents results on modeling on potential suitable habitats for the salamander Liua shihi under future global climate change. The salamander is endangered species; thus, such data are valuable for its protection. I found this manuscript interesting, however, I think it could be improved.

General remarks:
Authors analysed data for two future climate scenarios (SSP126, SSP585) of three periods 2021-2040, 2041-2060, 2061-2080. I am not sure, why Authors have chosen such periods – presently we have the year 2024, thus periods, for example, 2025-2045, 2046-2065, 2066-2080 would be better, I suppose.
However, if Authors will decide to left such periods, it should be explained. Additionally, I am not sure if the word “future” (for years “2021-2024”, e.g., lines 323-324 “Two periods in the future (2021-2040 and 2061-2080),…”) is the best one. 

I do not understand the last sentence of the Conclusion section (i.e., “Since only one species distribution model was used in this study, more models should be incorporated to obtain a higher prediction accuracy in the future.”). I agree that the results are not general for all, e.g., amphibian species. However, I think, that these models were used for more species, by many scientists(?)

The models predict distribution based on the predicted global climate change. However, for distribution of the species, human activity (e.g. destroying of habitat) could be crucial. I feel that more information in the area are necessary (in the Discussion section). See e.g. lines 95-96 “In recent years, the wild population of L. shihi is decreasing continuously due to human induced perturbations”.
Figure 7 Potentially suitable climatic distribution of Liua shihi… – if it possible that this species will migrate to other areas showed on the maps (with “Low”, and “Moderate” suitability)?

Figures
They could be improved. For example, Figure 1. What is showed on the small map (in the right low corner of the large map)? The same for Figure 7. (however, generally the Figure 7. is difficult to read – the maps are too small, I think).
“Figure 2. Correlation matrix between environmental variables.” All used abbreviation should be explained (in the legend, or e.g., as information added as supplementary materials).
Table 1. If all Units are correct? See, e.g., Mean UV-B of lowest month: “/”. Additionally, for me it is surprising, that the Permutation Importance of Annual mean UV-B is 0%, but for Mean UV-B of lowest month 13.2%. Check it please.

Generally, better captions of figures and tables would be useful for readers. In scientific papers, captions of figures and tables should be ‘self-explaining’, i.e., should provide sufficient information to the readers without looking for information in the text. Thus, the legends should be prepared in a better way, I believe.
See for example:
­– “Figure 3. Assessment metrics of MaxEnt generated by ENMeval.”
What is “RM”? Why on the y axis there is “Value” (value of what?)?
– “Table 2. Areas of suitable habitats for…”.
Why some values are presented with five decimal points, but other with four only? Generally, the table is not easy to read (specific values are not easy to compare). 

Several specific comments:
Line 167. “L, LQ, H, LQH, LQHP and LQHPT” – the abbreviations should be explained.
Line 197 “0.992 ± 0.002”. If “±” means ±SE (the standard error)? It should be precisely stated.
Line 228 “45.61×104 km2”. Please forgive me for my possible misunderstanding, but is it correct value? (see “104 km2”). The same for lines 232, 235, 237.
Lines 278, 291, 294 and several others: “L.shihi” – it should be “L. shihi” (i.e., with space).

Author Response

Reviewer 3:

I have read the manuscript entitled “Prediction of potential suitable distribution areas for an endangered salamander in China” by Jiacheng Tao and colleagues, submitted to the Animals magazine. The manuscript presents results on modeling on potential suitable habitats for the salamander Liua shihi under future global climate change. The salamander is endangered species; thus, such data are valuable for its protection. I found this manuscript interesting, however, I think it could be improved.

Reply: We appreciate the general interests of Reviewer 3 for our study. We have carefully and thoroughly revised this manuscript following Reviewers comments. Details have been provided below:

General remarks:

Authors analysed data for two future climate scenarios (SSP126, SSP585) of three periods 2021-2040, 2041-2060, 2061-2080. I am not sure, why Authors have chosen such periods – presently we have the year 2024, thus periods, for example, 2025-2045, 2046-2065, 2066-2080 would be better, I suppose.

Reply: We agree that the period of 2025-2045 could be better. However, the period of the climate variables layer is fixed based on the WorldClim website (https://www.worldclim.org/), and we cannot select the period randomly.

However, if Authors will decide to left such periods, it should be explained. Additionally, I am not sure if the word “future” (for years “2021-2024”, e.g., lines 323-324 “Two periods in the future (2021-2040 and 2061-2080),…”) is the best one.

Reply: These parts have been revised to make them clearer (L. 354).

I do not understand the last sentence of the Conclusion section (i.e., “Since only one species distribution model was used in this study, more models should be incorporated to obtain a higher prediction accuracy in the future.”). I agree that the results are not general for all, e.g., amphibian species. However, I think, that these models were used for more species, by many scientists(?)

Reply: This sentence has been revised to make it accurate (L. 385-387).

The models predict distribution based on the predicted global climate change. However, for distribution of the species, human activity (e.g. destroying of habitat) could be crucial. I feel that more information in the area are necessary (in the Discussion section). See e.g. lines 95-96 “In recent years, the wild population of L. shihi is decreasing continuously due to human induced perturbations”.

Reply: We totally agree that human activity could be an important factor that affect the distribution of L. shihi. However, since we only wanted to focused on the climate effects in the present study, the human activity could be another story in the near future.

Figure 7 Potentially suitable climatic distribution of Liua shihi… – if it possible that this species will migrate to other areas showed on the maps (with “Low”, and “Moderate” suitability)?

Reply: Based on the results, we can only say that these places are high, moderate, or low suitable for the living of L. shihi in the future. We are not sure this species can migrate to these places actively.

Figures

They could be improved. For example, Figure 1. What is showed on the small map (in the right low corner of the large map)? The same for Figure 7. (however, generally the Figure 7. is difficult to read – the maps are too small, I think).

Reply: The right corner of the map indicated the South China Sea islands and reefs, and it is typically presented like this in many previous studies (e.g., Aihua et al., 2023). For Figure 7, although it is small for each map, we can still find the distribution of suitable area in China. We have provided the name of each province in Figure 1, so that people can easily link the distribution and the provinces.

“Figure 2. Correlation matrix between environmental variables.” All used abbreviation should be explained (in the legend, or e.g., as information added as supplementary materials).

Reply: Done (L. 177).

Table 1. If all Units are correct? See, e.g., Mean UV-B of lowest month: “/”. Additionally, for me it is surprising, that the Permutation Importance of Annual mean UV-B is 0%, but for Mean UV-B of lowest month 13.2%. Check it please.

Reply: Table 1 has been revised to make it more accurate (L. 153-154).

Generally, better captions of figures and tables would be useful for readers. In scientific papers, captions of figures and tables should be ‘self-explaining’, i.e., should provide sufficient information to the readers without looking for information in the text. Thus, the legends should be prepared in a better way, I believe.

See for example:

­– “Figure 3. Assessment metrics of MaxEnt generated by ENMeval.”

What is “RM”? Why on the y axis there is “Value” (value of what?)?

– “Table 2. Areas of suitable habitats for…”.

Why some values are presented with five decimal points, but other with four only? Generally, the table is not easy to read (specific values are not easy to compare).

Reply: All the Tables and Figures have been carefully revised.

Several specific comments:

Line 167. “L, LQ, H, LQH, LQHP and LQHPT” – the abbreviations should be explained.

Reply: Done (L. 185-186).

Line 197 “0.992 ± 0.002”. If “±” means ±SE (the standard error)? It should be precisely stated.

Reply: Done (L. 121-123).

Line 228 “45.61×104 km2”. Please forgive me for my possible misunderstanding, but is it correct value? (see “104 km2”). The same for lines 232, 235, 237.

Reply: We apologize for this mistake. These parts have been carefully revised (L. 255-295).

Lines 278, 291, 294 and several others: “L.shihi” – it should be “L. shihi” (i.e., with space).

Reply: Done (L. 303, 320, 323, 333, 368, 371).

References:

Aihua F, Erhu G, Xiaoping T, Zengli L, Faxiang H, Zhenjie Z, Jiadong W, Xiaofeng L. MaxEnt Modeling for Predicting the Potential Wintering Distribution of Eurasian Spoonbill (Platalea leucorodia leucorodia) under Climate Change in China. ANIMALS-BASEL 2023;13(5).

Round 2

Reviewer 1 Report

Comments and Suggestions for Authors

Ms has been revised and improved well, please accept the ms. However, please modify figure 6, it is not clear.